# Assessing Microplastics and Nanoparticles in the Surface Seawater of Venice Lagoon—Part I: Methodology of Research

**DOI:** 10.3390/ma17081759

**Published:** 2024-04-11

**Authors:** Teresa Cecchi, Davide Poletto, Andrei Constantin Berbecaru, Elfrida Mihaela Cârstea, Maria Râpă

**Affiliations:** 1Chemistry Department, Istituto Technico Technologico, Via Montani 7, 63900 Fermo, Italy; cecchi.teresa@istitutomontani.edu.it; 2Venice Lagoon Plastic Free, Castello 2641, 30122 Venice, Italy; 3Faculty of Materials Science and Engineering, National University of Science and Technology Politehnica Bucharest, 060042 Bucharest, Romania; andrei.berbecaru@upb.ro; 4National Institute of R&D for Optoelectronics INOE 2000, Atomistilor 409, 077125 Magurele, Romania; elfrida.carstea@inoe.ro

**Keywords:** microplastics, nanoparticles, surface water, Venice Lagoon

## Abstract

Microplastics (MPs) and nanoplastics (NPs) both represent significant concerns in environmental sciences. This paper aims to develop a convenient and efficient methodology for the detection and measurement of MPs and nanoparticles from surface seawater and to apply it to the water samples collected from the UNESCO site of Venice and its lagoon, more precisely in the Venice-Lido Port Inlet, Grand Canal under Rialto Bridge, and Saint Marc basin. In this study, MPs were analyzed through optical microscopy for their relative abundance and characterized based on their color, shape, and size classes, while the concentration and the mean of nanoparticles were estimated via the Nanoparticle Tracking Analysis technique. Bulk seawater sampling, combined with filtration through a cascade of stainless-steel sieves and subsequent digestion, facilitates the detection of MPs of relatively small sizes (size classes distribution: >1 mm, 1000–250 μm, 250–125 μm, 125–90 μm, and 90–32 μm), similar to the size of MPs ingested by marine invertebrates and fishes. A protocol for minimizing interference from non-plastic nanoparticles through evaporation, digestion, and filtration processes was proposed to enrich the sample for NPs. The findings contribute to the understanding of the extent and characteristics of MPs and nanoparticle pollution in the Venice Lagoon seawater, highlighting the potential environmental risks associated with these pollutants and the need for coordinated approaches to mitigate them. This article is based on scientific research carried out within the framework of the H2020 In-No-Plastic—Innovative approaches towards prevention, removal and reuse of marine plastic litter project (G.A. ID no. 101000612).

## 1. Introduction

Today’s escalating demand for plastic products, coupled with insufficient recycling efforts, has led to a significant buildup of plastic waste as these items reach the end of their life cycle. It is estimated that about 500 kilotons of plastic per year enter the world’s oceans [1]. Marine plastic litter can be found stranded on the beach, constituting 50–90% of sampled items [2,3] as well as floating [4,5] or submerged in marine environments [6]. In aquatic environments, floating plastic waste can break down, in the long term, into smaller fragments, specifically micro-(MPs) and nanoplastics (NPs), due to sunlight exposure, named photoaging [7]. This process occurs gradually over time as they are exposed to sunlight and environmental stressors such as waves and temperature fluctuations. These small fragments are commonly less than 5 mm in size and range from a few nanometers to micrometers, referred to as MPs and NPs, which are considered emerging pollutants. MPs result via the fragmentation of larger plastics, while NPs result from the further breakdown or progressive fragmentation of MPs. Larger plastics exhibit a lower rate of photoaging compared to smaller plastics. The reason for the discrepancy between larger plastics and a lower rate of photoaging compared to smaller plastics is that the nanoplastics exhibit a specific surface area, which enhances the generation of reactive oxygen species (ROS) [8] having a higher capacity of forming hydroxyl radicals (HO·) [7]. The degradation of plastic debris under sunlight exposure leads to an increase in surface roughness, allowing smaller particles to easily leach additives from their composition and adsorb other chemical toxic pollutants from aquatic media, making them more dangerous to the aquatic environment. ROS are involved in the degradation of plastic debris under sunlight exposure. The release of MPs and NPs can also occur during their production and use. For instance, tire abrasion, release of fibers during washing, and plastic bottle single-use products are notable examples of MPs release during the use phase [9,10,11,12]. Once released, these particles can contaminate soils, freshwater ecosystems, drinking water [13], and marine environments [14], and can be ingested by aquatic organisms such as marine algae, crabs, and whales. It was reported that over 701 marine species ingested MPs [15]. Many studies have demonstrated the long persistence and abundance of MPs [16,17], their unique reactivity and bioavailability for aquatic organisms [18], binding and absorbing of other chemical pollutants from aqueous media [19,20,21,22,23,24,25], and, thus, increasing the potential risk to ecosystems and human population. The estimate of 1.2 million tons of plastic stock that has accumulated in the Mediterranean Sea underscores the severity of plastic pollution [3]. They can disrupt the ecological balance, affect biodiversity, and contribute to the degradation of habitats. The potential implications of plastic particles to both natural ecosystems and human health are not fully understood. Only a few studies, reporting on NPs, involve test models of nanoparticles, artificial seawater, and rarely coastal seawater [26,27,28].

Table 1 reveals some features of MPs and NPs detected in surface water environments.

The results show heterogeneity regarding the occurrence of MPs, their categorization, and color. Floating plastic objects can be collected from water bodies using various methods, such as neuston manta, bottles, buckets, jerry cans, or drones [29,31,34,35]. The assessment of MPs demonstrated the variations in the abundance due to the rainfall, hurricanes, and seasons [2,3,25,34]. For example, it was reported that in Banderas Bay, in Mexico, the higher MP mass occurred one month after the first peak of precipitation [31]. In another case [35], it was found that the season did not affect MP occurrence. Multiple analytical techniques based on mass or particle size determinations are usually required to obtain information about the concentration, chemical composition, and shape of MPs and NPs. MP identification techniques include microscopic observations [36], µ-Raman [37,38], Fourier transform infrared spectroscopy (FTIR) [38,39], µ-FTIR [31], scanning electron microscopy (SEM)/energy dispersive X-ray spectroscopy (EDX) [36,38], and micro-optofluidic platform [36,40], while the NPs have been evaluated using thermal desorption-proton transfer reaction-mass spectrometry (TD-PTR-MS) [29,41], dynamic light scattering (DLS) [26,42,43], transmission electron microscopy (TEM) [44,45], surface-enhanced Raman spectroscopy (SERS) [46], and pyrolysis-gas chromatography/mass spectrometry Py-GC/MS analytical techniques [47]. Since no single analytical method is able to obtain the physical (size, shape, and color) and chemical (polymer type) characteristics of MPs in a single step, the combination of several parallel approaches should be applied and considered. The most identified polymers in marine environments are PP and PE [31,34,48], PET [49], PS [50], and alkyd resins from marine coatings [32], as these are the most massively produced polymers worldwide.

Despite considerable research endeavors into MPs and NPs, numerous gaps in knowledge persist, especially regarding the absence of standardized methodologies for evaluating and quantifying these pollutants and their health impacts. This challenge hampers the ability to compare results across studies, underscoring the need for enhanced research techniques. The progress of research on standardized methods for assessing and quantifying these pollutants and their health effects has significantly advanced over the years, driven by the increasing recognition of the impact of pollutants on human health and the environment. The key advancements in this field are illustrated in reports elaborated by the Joint Group of Experts on the Scientific Aspects of Marine Environmental Protection (GESAMP) [51,52], the Joint Research Center (JRC) [53], and the National Oceanic and Atmospheric Administration’s (NOAA) [54], which aim to standardize methods for sampling, analysis, and reporting of MPs data. Also, interlaboratory comparison studies were organized with the purpose of predetermining the recovery rates of spiked MPs in wastewater and sludge samples [55], identifying the type and quantity of MPs [56], generating MPs in standard sample bottles [57], and developing reference materials for method validation in microplastic analysis [58].

This paper aims to develop a convenient and efficient methodology for the detection and measurement of MPs (sized 1→1000 µm) and nanoparticles (<1 µm) from aquatic environments and to apply it to seawater samples collected from the Venice Lagoon. As nanoparticles, in addition to NPs, dissolved fractions including exopolymeric substances, pedogenic substances such as fulvic and humic acids, polysaccharides, proteins, clay minerals, inorganic oxides, and marine snow are found in aquatic environments [59,60]. In this analysis, MPs were assessed for their relative abundance and characterized based on color, shape, and size classes, while the concentration and mean of nanoparticles were estimated. Additionally, a mass quantification approach for nano- and MPs particles detected in seawater samples is proposed.

## 2. Materials and Methods

### 2.1. Sampling Procedure

Five and a half liters of seawater were collected via bulk method on 6th December 2022 from three locations across Venice Lagoon, Italy: Venice-Lido port inlet, Grand Canal under Rialto Bridge, and Saint Marc basin, at ~30 cm below the surface employing commercially unused plastic bottles. The geographical coordinates for each sampling location are shown in Figure 1. All sample containers with a tight-fitting lid were previously rinsed three times with seawater, before use to remove any potential contaminants. The sampling procedure was undertaken in the following environmental conditions: temperature of 7 °C, Beaufort number 2, wind speed of 5.6 mph, wind direction NE, humidity 99%, and atmospheric pressure of 30 mmHg. The outside of the closed sample bottles was rinsed with distilled water several times prior to any further manipulations. This step removes contaminations, which could possibly occur during shipping. The samples were stored at 4 °C prior to analysis, while particular attention was paid to avoid contamination of the samples.

With a surface area of about 550 km^2^, the Venice Lagoon is one of the largest lagoons in the Mediterranean region, providing important ecosystem services. It is situated along the Adriatic coast of northeastern Italy and encompasses the cities of Venice and Chioggia, as well as several smaller towns and islands. Venice Lagoon is a UNESCO World Heritage Site, an important tourist destination, and serves as a buffer against coastal erosion providing water filtration and carbon sequestration.

### 2.2. Processing of MPs and Nanoparticles

Figure 2 shows the research methodology for the estimation of MPs and nanoparticles from seawater samples.

#### 2.2.1. Mitigation of Contamination

The personnel wore 100% cotton protective cloths and nitrile gloves. Glassware, sieves, and working surfaces were cleaned with 70% ethanol and then rinsed with ultrapure water to prevent cross-contamination. All beakers, graded cylinders, and tips for the micropipette were rinsed three times with filtered ultrapure water. Ultrapure water, ethanol, the solution of 30% hydrogen peroxide (H_2_O_2_), and saline solution were filtered by GF6 filter inorganic binder, 47 mm diameter, 1 µm pore size (Hahnemühle, Dassen, Germany) prior to use. For nanoparticle assessment, all reagents were filtered by mixed cellulose esters sterile filters, 47 mm diameter, 0.22 µm pore size (Millipore, Molsheim, France) prior to use. Glasswares containing samples were covered with aluminum foil for transportation to the oven and analytical balance. All windows and doors were closed in order to minimize the air circulation.

#### 2.2.2. Wet Sieving

A stack of successive stainless-steel mesh sieves (Test sieves, ISO 3310-1 [61], Fritsch GmbH, Idar-Oberstein, Germany) with mesh sizes of 1 mm, 250 μm, 125 μm, 90 μm, and 32 μm was used to filter the samples. Five corresponding retentates (R1) and one filtrate (F1) were collected from all mesh sieves. The retained solid particles on the stainless-steel sieves were rinsed with filtered ultrapure water and carefully transferred into glass beakers. The content of each beaker was evaporated in an oven at a temperature below 70 °C (to avoid the MP’s destruction).

#### 2.2.3. Digestion with H_2_O_2_ and Density Separation

The retentates were subjected to a digestion step, at room temperature, to destroy the organic matter, using 20 mL of 30% (*w*/*v*) H_2_O_2_, an environmentally benign oxidant, until no foam was observed. The reagent, previously filtered with a filter of 0.22 µm pore size, was used to preserve the plastic, while removing organic material. Biological material is often confused with plastics (e.g., darker algae fragments), leading to overestimation of environmental concentrations and increasing the number of particles subjected to further analysis. The oxidizing agent is able to digest organic matter more efficiently than NaOH and HCl, with little to no degradation of possible polymers. It was reported that the HO· is very useful in removing natural organic matter, having the advantage of being non-selective [62,63].

Density separation using a NaCl solution with a density of 1.2 g/cm^3^ was an effective method for separating MPs via flotation. The principle behind this method is that objects with a lower density than the solution will float, while those with a higher density will sink.

Subsequently, each resulting mixture is subjected to vacuum ultrafiltration (Mini Diaphragm Vacuum Pump Laboport^®^N 96, KNF Japan Co., Ltd., Tokyo, Japan, 0.30–0.42 m^3^/h, <130 mbar abs.) equipped with a suction filtration unit and Whatman^®^glass microfiber filters, Sigma-Aldrich, St. Louis, MI, USA, binder-free (Grade GF/B 47 mm, 1 μm pore sizes). The retentates were finally rinsed with 70% ethanol in each step. Five filters were collected for each retentate and were stored in labeled closed Petri dishes (PDR1), dried at room temperature, and kept in a desiccator until subsequent analysis.

#### 2.2.4. Optical Microscopy

The MPs isolated from the seawater samples were examined via the optical microscope, Olympus Stream Essential 1.9.3 Software (OLYMPUS BX 51 M, Olympus Corporation, Tokio, Japan) with magnification 50× up to 1000×. The microscope is equipped with a color digital camera, min resolution of 3 MP, and a CCD sensor of min ½” through the determination of size, color, and morphology. Optical microscopy stands out as the optimal technique for the prompt identification of MPs, thanks to its capacity to easily differentiate plastic materials from other substances such as fats, mineral particles, and cellulose fibers [64]. To avoid misidentification and underestimation of MPs (>1 mm), several criteria are applied to optical investigation [65,66]:▪The particles have no observable organic or cellular structures;▪In the case of fibers, the diameter should be consistent along their length;▪Particles should present clear and homogeneous colors;▪The further high magnification should be used in the case of transparent or white particles.

Triplicate samples of ultrapure water, filtered with a 0.22 µm pore size filter and passed all steps applied for analyzing the Venice Lagoon water samples, were used as negative control (*n* = 3).

#### 2.2.5. Scanning Electron Microscopy/X-ray Energy Dispersive Spectrometry

SEM coupled with EDX (FEI, QUANTA 450 FEG, Eindhoven, The Netherlands) images were acquired in high vacuum (HV) working mode, at 30 kV, using backscattered electron (CBS) and secondary electron (SE) detectors for the inspection of the morphology of samples.

#### 2.2.6. Filtrate Treatment

The filtrates (F1) resulting after the cascade filtration step were stored in glass bottles and used for the determination of chloride content (g/L), salinity, electrical conductivity (EC) (mS/cm), pH, and total dissolved solids (TDS) (g/L), as well as estimation of nanoparticles. The pH was assessed with a CONSORT C831 multi-parameter analyzer, while the TDS, salinity, and EC parameters of each water sample were measured with a Consort C862-multi-parameter analyzer. The amount of chloride was estimated via titration method using the Mohr Method.

Nanoparticles’ assessment involves the filtration of 20 mL of filtrate (F1) via a binder-free glass microfiber filter, with a pore size of 1 µm. The obtained filtrate (F2) was poured into a glass beaker and evaporated at a temperature below 70 °C. The residue was submitted to a digestion step to destroy the organic matter, by using 20 mL of 30% (*w*/*v*) H_2_O_2_, at room temperature. After no reaction was observed (approx. 5 days), the residue was evaporated again at a temperature around 70 °C. After evaporation, approx. 20 mL of negative control (ultrapure water) was added to a beaker. This solution was used for measuring the concentration of nanoparticles (particle/mL), mean size dimension, and particle size distribution, via the Nanoparticle Tracking Analysis (NTA) tool, NTA 3.2 Dev Build 3.2.16 (Nanosight NS300, Malvern Panalytical, Worcestershire, UK). The NTA method was used to characterize particles with sizes between 10 nm and 1000 nm. The Nanosight NS300 instrument is equipped with a laser emitting at 532 nm and a sCMOS camera recording at 60 fps. The temperature controller was set at 25 °C. Five measurements of 60 s were recorded for each sample. The hydrodynamic diameter (*d_h_*) of the particles tracked is calculated by applying the two-dimensional Stoke–Einstein equation [67]:(1)$$d_{h}=\frac{K_{B}Tt_{s}}{3\pi \eta D}$$
where *K_B_* is the Boltzmann’s constant, *T* is the absolute temperature, *t_s_* is the measurement time, *η* is the viscosity, and *D* is the diffusion coefficient.

Our approach involved removing organic matter from a water sample, evaporating the liquid, and then adding an equal amount of ultrapure water with the same volume as that of the original sample. The intention was to eliminate all organic interferents. By subtracting the concentration of nanoparticles measured in the ultrapure water (treated in the same manner as the water sample) from the processed water sample, theoretically, the neat concentration of nanoparticles in the processed samples should be obtained.

For positive control in nanoparticle analysis, polystyrene nanoparticles (PS NPs) (standard latex NTA4089, Malvern Panalytical, Worcestershire, UK, certified diameter 204 ± 3.1 nm) were diluted 20× in medical grade saline solution of 9 mg/mL NaCl.

After the treatment, each water sample was analyzed in triplicate, in five repetitions.

#### 2.2.7. Mass Estimation

The total weight of MPs and nanoparticles (*M*) in µg was estimated based on the polymer density and volume of every particle, with an assumption that all the particles are spherical with visible diameter, and the density of all particles was estimated to be 1 g/cm^3^, according to Equation (2).
(2)$$M=\rho \sum_{i=1}^{n}C_{i}\frac{1}{6}\pi d_{i}^{3},\;\mu g$$
where *ρ* is the density (g/cm^3^), *C_i_* is the number of particles, and *d_i_* is the diameter of each particle.

The diameter of particles was considered to be the mesh of the sieve, in the case of MPs detected via optical microscopy, and the mean size of particles in the case of NPs detected via NTA.

#### 2.2.8. Statistical Analysis

Statistical analysis was performed using Microsoft Excel, Office 2021, version 2403. ANOVA (analysis of variance) was used to analyze differences in MP characteristics from different sites. For all statistical analyses, *p* values < 0.05 were considered to be significant.

## 3. Results

### 3.1. Physical-Chemical Parameters

The results of the physical and chemical analyses of water samples are shown in Table 2.

The pH value of seawater ranged between 7.7 and 7.9, making it slightly alkaline. MPs in seawater can interact with dissolved ions and organic matter, which can affect their aggregation, sinking, or floating behavior. The electrical conductivity of seawater is related to the concentration of dissolved salts, primarily sodium chloride (NaCl), which increases the conductivity. The presence of dissolved ions can lead to the formation of ion complexes around MPs, altering their surface charge.

### 3.2. Microplastic Evaluation

#### 3.2.1. Optical Microscopy

Some images of MPs detected through optical microscopy in the Venice seawaters are shown in Appendix A. Concentrations of 2.6 ± 0.83 MPs/L, 2 ± 1.01 MPs/L, and 1.57 ± 0.91 MPs/L were detected for Venice-Lido port inlet, Rialto Bridge, and Saint Marc water samples, respectively (Figure 3a). No MPs collected from 32 µm to 1 mm mesh sieves were found in negative control. The variance in the occurrence of MPs among the three sites could be associated with the circulation and types of currents prevalent in the Venice Lagoon, described as a large-scale cyclonic meander [68]. Thus, the Venice-Lido port inlet area showed higher MP abundance compared with Saint Marc and Rialto Bride. It is possible that the stronger currents from the Venice-Lido port inlet act as traps for MPs compared to the Rialto Bridge and Saint Marc sites, while weaker currents, may potentially facilitate the export of MPs. Baini et al. [35] also suggested an increase in MP concentration from coastal areas to distant waters. Other authors [25] emphasized the potential role of harbors in contributing to MPs in marine environments. We found no statistically significant differences in the MP abundance between the sites (*p* > 0.05). Among all three sites, the average abundance in the Venice Lagoon was estimated to be 2.06 ± 1.59 MPs/L.

The size class distribution revealed a marked prevalence for particles in the range of 125–250 µm for Venice-Lido port inlet (41.8%), Rialto Bridge (54.5%) and Saint Marc basin (53.8%) (*p* < 0.05), followed by particles > 1 mm for Saint Marc basin (30.7%) (*p* < 0.05), and particles ranging from 90 to 125 µm for Venice-Lido port inlet (27.9%) (*p* < 0.05) (Figure 3b). Approximately 15% of MPs are situated in the 32–90 µm class size (*p* < 0.05) for Venice Port Inlet and Rialto Bridge. Our predominantly isolated MP sizes in the surface seawaters are in line with other reported data, for example, 100–200 μm for the South China Sea and the western Pacific Ocean [69], and 50 μm–300 μm for the Whitsunday Islands region [70].

The predominant shape of MPs is filament or fiber. Fragment particles accounted for 3% of Rialto Bridge and 19% of the Saint Marc basin—Figure 3c.

Seven chromatic components were detected in stacked bar charts, with black, colorless, and blue being predominant at Venice-Lido port inlet (Figure 4a). For Rialto Bridge, the dominant colors were black, red, and yellow (Figure 4b), while blue and black were primarily found at the Saint Marc basin sampling location (Figure 4c). Similar colors of MPs have been reported in other studies [48,71,72].

#### 3.2.2. SEM/EDX Analysis

Scanning electron microscopy (SEM) and energy-dispersive X-ray spectroscopy (EDX) were employed for the morphological investigation of potential MPs and the identification of their elemental composition. The SEM/EDX screening method utilized surface morphology and elemental composition to assess whether each particle had the potential to be plastic. We specifically looked for spectra indicating a significant concentration of carbon, as these were considered potential candidates for MPs.

Figure 5a–f shows the images for possible MPs and negative control detected via SEM analysis.

SEM analysis showed fibers with dimensions of 266 µm and 513 µm for Venice-Lido port inlet (Figure 5a,b), 222 µm for Rialto Bridge (Figure 5c), and two fragments with lengths of 316 µm and 197 µm for Saint Marc basin (Figure 5e).

The patterns and elemental compositions of micro-fragments detected in seawaters are shown in Figure 6a–f and Table 3.

The primary constituents for micro-fragments detected in Venice-Lido port inlet (Figure 6a,b) and Grand Canal at the Rialto Bridge water samples (Figure 6c) are carbon 16.45–62.83 wt. % and oxygen 32.51–47.78 wt.%. The micro-fragments detected in the Saint Marc basin water sample (Figure 6e,f) are characterized by 31.01–41.85 wt.% carbon, 42.32–39.28 wt.% oxygen, and 3.69–9.83 wt.% silicon.

### 3.3. Estimation of Size and Concentration of Nanoparticles

Figure 7a shows the size distribution profile (SD) of particle within the samples, along with the concentration of particles per mL determined from the NTA analysis. The concentration of particles/mL plotted in Figure 7b was calculated based on all particles present in the water sample and oxidant agent, after subtracting the calculated concentration of particles in the negative control.

The profiles of concentration intensity as a function of the size of nanoparticles detected using NTA are shown in Appendix A. The mean size of NPs decreased in the following order: Saint Marc basin (484 nm) > Grand Canal at the Rialto Bridge (387 nm) (*p* < 0.01) > Venice-Lido port inlet (325 nm)—Figure 7a. The concentration of nanoparticles (ranging from 3.71 × 10^7^ particles/mL (*p* < 0.05) to 1.22 × 10^8^ particles/mL (*p* < 0.001)) follows the same trend with their size—Figure 7b. A total of 10% of particles (D10) have sizes in the range of 202–341 nm, while 90% of particles (D90) are situated in the range of 443–592 nm, with the highest values recorded for the Saint Marc water sample.

### 3.4. Estimation of MP and Nanoparticle Concentration

The estimation of MPs and nanoparticles’ concentration depends on the number of particles and their diameters. 

Table 4 shows the concentration of MP and nanoparticle of samples reported as mg to 1 m^3^ of water samples. According to our results, although the occurrence of MPs is less predominant in the Saint Marc basin site, the calculated concentration is higher for this location due to the high number of MPs detected on the 1 mm mesh sieve.

The estimated concentration of nanoparticles follows the same trend as the provided data using NTA, owing to the insignificant variation between the mean diameters of the recorded nanoparticles. A concentration of PS NPs of 4.2 mg/m^3^ was reported for the Wadden Sea site, using the TD-PTR-MS method [29].

## 4. Discussion

A protocol was proposed for detecting microplastics (MPs) and nanoparticles smaller than 1 mm in surface seawater, applied at three Venice Lagoon locations. The processing of MPs and nanoparticles varies based on the sampling method and factors such as temperature, electrical conductivity, pH, salinity, and total dissolved substances.

Although plankton nets are valuable tools for studying marine ecosystems, they may not capture the full range of MPs, particularly smaller particles. For instance, Covernton et al. [73] affirmed that the plankton nets may underestimate MP concentrations by up to four orders of magnitude compared to bulk seawater samples. This is because larger mesh sizes are less effective in capturing smaller MPs and NPs, leading to lower reported concentrations. In addition, zooplankton or small fish typically feed on particles in the size range of micrometers, so sampling methods with smaller mesh sizes (<100 μm) are required to accurately quantify the MPs they ingest. More than 80% of the MPs found in seawater, marine sediment, and corals were smaller than 2 mm [74]. The majority of MPs ingested by marine invertebrates and fishes are often smaller than 300 μm in diameter [75,76,77]. The bulk technique used in this study allowed the detection of small MP size (size classes’ distribution: >1 mm, 1000–250 μm, 250–125 μm, 125–90 μm, and 90–32 μm). This classification enables the assessment of the distribution of MPs in the environment, their potential for ingestion by aquatic organisms, and their ecological and health implications.

Zooplankton with sizes ranging between 200 µm to 200 mm, have the capacity to potentially ingest suitably sized MPs [78], often mistaking them for food in aqueous ecosystems. Zooplankton, such as copepods, krill, and various larval forms of marine animals, are also used as indicators of microplastic pollution in aquatic ecosystems by ingesting microplastic particles present in that environment. They help quantify the level of microplastic contamination in aquatic environments by measuring the number of MPs ingested by zooplankton. Ecotoxicological studies assess how microplastic ingestion affects zooplankton survival, reproduction, growth, and physiology. Also, model studies are conducted with zooplankton samples to further understand the dynamics of microplastic pollution in aquatic ecosystems [79]. Various organisms in aquatic ecosystems, including copepods, mysid shrimps, rotifers, cladocerans, ciliates, and polychaete larvae, are a significant part of the food chain and are often food sources for other organisms, including fish, birds, and marine mammals. The size of MPs is a significant factor influencing the frequency and quantity of plastics ingested by these organisms. Bermudez and Swarzenski [78] investigated the size relationship between predators and prey in aqueous ecosystems. Thus, for dinoflagellates, other flagellates, ciliates, rotifers and copepods, and cladocerans and meroplankton larvae, the size ratios with their prey are 1:1, 3:1, 8:1, 18:1, and approximately 50:1, respectively. In addition, the study notes a difference in prey selectivity between filter feeders and raptorial-interception feeders. Filter feeders tend to prefer relatively smaller prey, while raptorial-interception feeders favor larger prey. In a study conducted by Sfriso et al. [80], it was reported that the size of MPs detected from macrophytes, as environmental indicators and pollutant bioaccumulators, in the Venice Lagoon predominantly falls within a range of Feret diameters of 210 µm. However, the planktonic web is not able to capture the full range of MPs, especially the smaller particles. It was reported that planktonic copepods, which show a range of prey sizes from 200 µm to 2000 µm, can reject 80% of the MPs due to the feeding-current and sensorial mechanism, independent of the type of polymer, morphology, and presence of biofilms [77]. Also, the residence time for the MPs in copepods is short, and them being swiftly expelled through fecal pellets.

The abundance of MPs in this study is correlated with the physical properties of seawater from coastal areas, such as salinity and total dissolved solids variations. Salinity refers to the concentration of dissolved salts in seawater, primarily composed of sodium chloride (NaCl) but also containing other ions such as magnesium, calcium, and potassium. These components affect the density of seawater, which in turn can influence the buoyancy of MPs. As salinity increases, the concentration of dissolved salts in the water increases and more MPs will float on the surface of the water. The highest values for salinity were recorded for the Venice-Lido port inlet water sample. These are well correlated with the high abundance of MPs for this site (1.57–2.6 MPs/L). Higher salinity can lead to an increase in the buoyancy of MPs, depending on factors, such as the size, shape, and composition of the particles. This finding is also supported by Jiang et al. [34]. It was considered that the plastic bottles did not contribute to the generation of MPs in water samples.

Also, it was observed that the abundance of MPs was inversely proportional to the concentration of nanoparticles measured in the Venice Lagoon. The higher number of nanoparticles/mL recorded in the case of the Saint Marc basin seawater sample signifies a high fragmentation rate of MPs. Other reported data showed a calculated concentration of 1.03 × 10^9^ NPs/L of seawater, detected by using TD-PTR-MS [29].

The EDX composition of micro-fragments from Figure 6a–c,e,f are compatible with the chemistry of common plastic marine litter. A higher amount of oxygen may be related to PET [36], while higher amounts of Si could suggest the presence of silicon-based MPs even if the silicon from the glass fiber filter may interfere. The object with the highest amount of silicon (33.07 wt.%) and no carbon is a diatom (Figure 6d). Diatoms, also known as Bacillariophyceae, have a significant contribution to aquatic ecosystems by converting dissolved carbon dioxide into oxygen [81,82].

Our estimated concentrations for MPs and nanoparticles were lower than other reported data, possibly due to the sampling method, detection technique used, and methodology applied. For instance, the MP weight concentrations were in the range of 0.6 μg/m^3^ in the North Atlantic and Siberian Arctic to 7.5 μg/m^3^ in the Barents Sea, when a ship-board underway pump-through system was used for sampling [83]. Also, concentrations from 5 µg/m^3^ to 428 µg/m^3^ for particle sizes between 0.3 mm to 5 mm were found in the Port of Gdynia, Baltic Sea [48].

The plastic pollution in the Venice Lagoon originated from both land-based and sea-based sources. Increased tourism in Venice, along with the consumption of single-use plastics such as water bottles, food containers, and packaging, as well as tourism-related activities such as boat tours, cruises, and recreational activities, can significantly contribute to plastic waste entering the Venice Lagoon. Additionally, abandoned, lost, or discarded fishing gear (32%) found in the marine debris from Venice [84,85] could be another significant source of microplastic pollution.

The limitations of this methodology are attributed to the rejection of MPs due to mesh sizes ranging from 32 μm down to 1 μm, potential interferences from co-existing colloids and fragments during the NTA technique [86], and the high probability of missing small and transparent MPs during the optical microscopy analysis. However, the detection and identification of MPs and NPs remain labor- and cost-intensive, requiring the achievement of high-sample throughput analysis as well as a standardized sampling plan and protocol.

Understanding the abundance and estimating the concentration of MPs and nanoparticles could help to assess the potential health risks both to the aquatic ecosystems and humans as well as to develop strategies to reduce plastic inputs.

## 5. Conclusions

These data represent an attempt to apply a simple methodology for quantifying small-sized MPs and nanoparticles that float in the Venice Lagoon.

Bulk seawater samples coupled with digestion and filtration using a cascade configuration of stainless-steel sieves enabled the detection of small mesh size MPs (size distribution > 1 mm, 1000–250 μm, 250–125 μm, 125–90 μm, and 90–32 μm).

The average abundance of MPs in the Venice Lagoon including all three sites, was 2.06 ± 1.59 MPs/L. The predominant colors of MPs are black, blue, yellow, red, colorless, green, and brown.

The protocol applied for removing or reducing the interference from non-plastic nanoparticles by using evaporation, digestion, and filtering showed nanoparticle concentrations for Venice Lagoon ranging from 3.71 × 10^7^ particles/mL to 1.22 × 10^8^ particles/mL.

Future work will be devoted to the investigation of the influence of glass bottles versus plastic bottles on the estimation of MPs and nanoparticles from seawaters.

## Figures and Tables

**Figure 1 materials-17-01759-f001:**
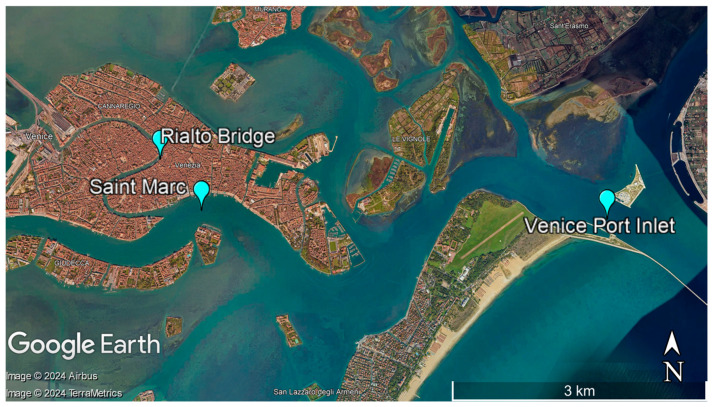
Sampling locations: Venice-Lido port inlet, GPS coordinates: latitude: 45.431508, longitude: 12.406952; Grand Canal under Rialto Bridge, GPS coordinates: latitude: 45.438350, longitude: 12.336311; and Saint Marc basin, GPS coordinates: latitude: 45.431962, longitude: 12.340953.

**Figure 2 materials-17-01759-f002:**
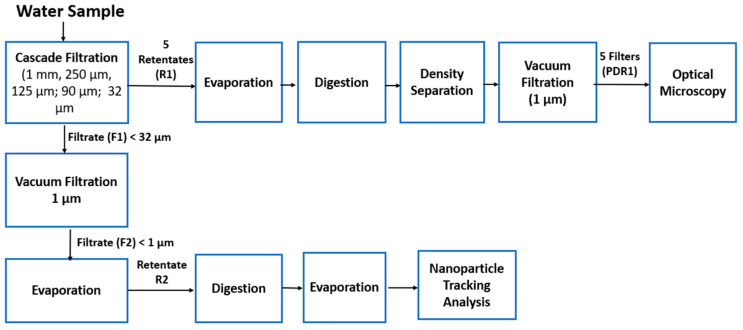
The methodology of research.

**Figure 3 materials-17-01759-f003:**
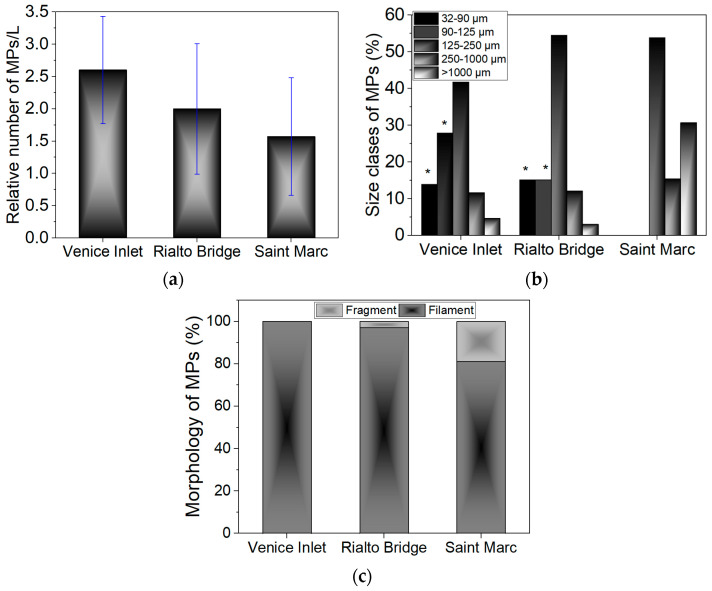
Potential abundance (%) of MPs detected on glass microfiber filters after digestion and filtration of seawaters and classified according to (**a**) abundance (*p* > 0.05), (**b**) distribution on size classes, and (**c**) shape (*p* > 0.05). * *p* < 0.05.

**Figure 4 materials-17-01759-f004:**
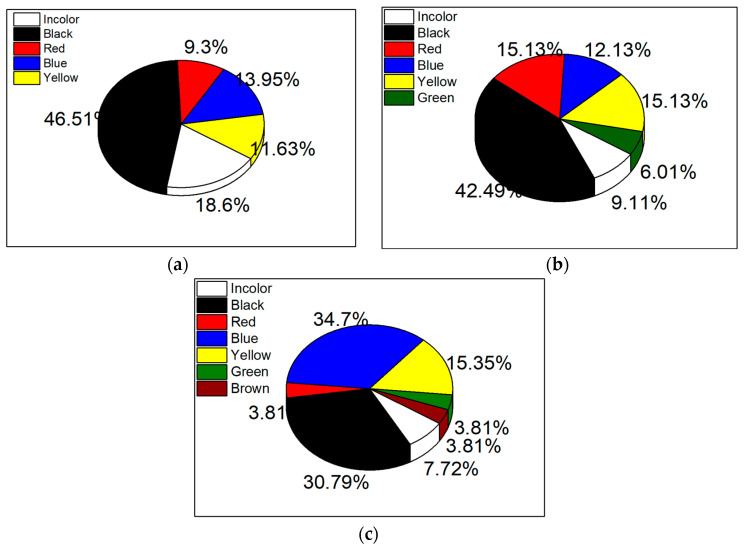
The chromatic constituents for MPs detected in (**a**) Venice-Lido port inlet, (**b**) Grand Canal at the Rialto Bridge, and (**c**) Saint Marc basin. For this test *p* > 0.05.

**Figure 5 materials-17-01759-f005:**
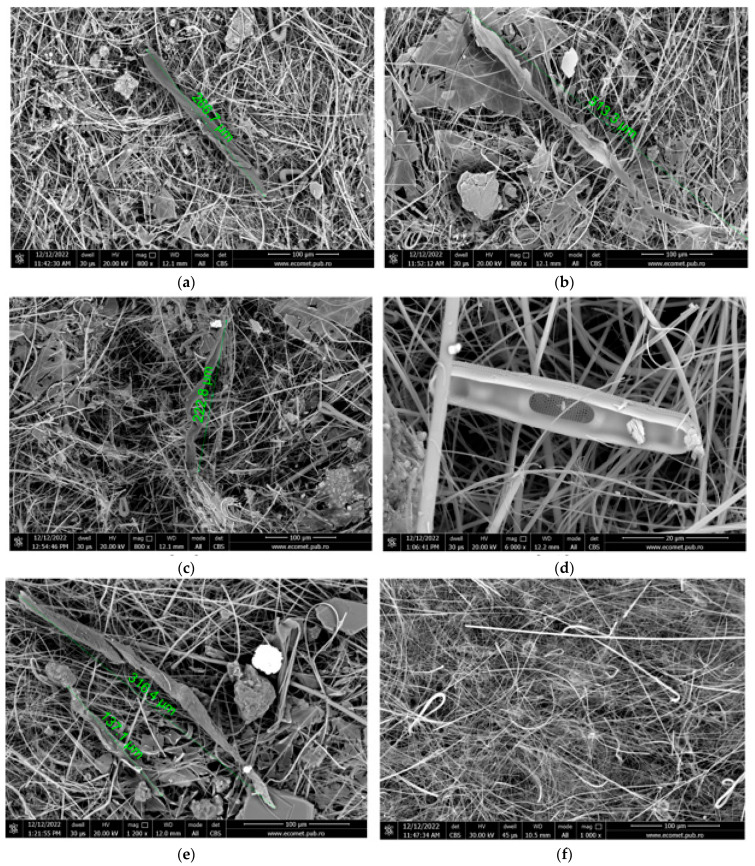
SEM images for investigated seawaters. (**a**,**b**) Venice-Lido port inlet, (**c**,**d**) Grand Canal at the Rialto Bridge, (**e**) Saint Marc basin, and (**f**) negative control.

**Figure 6 materials-17-01759-f006:**
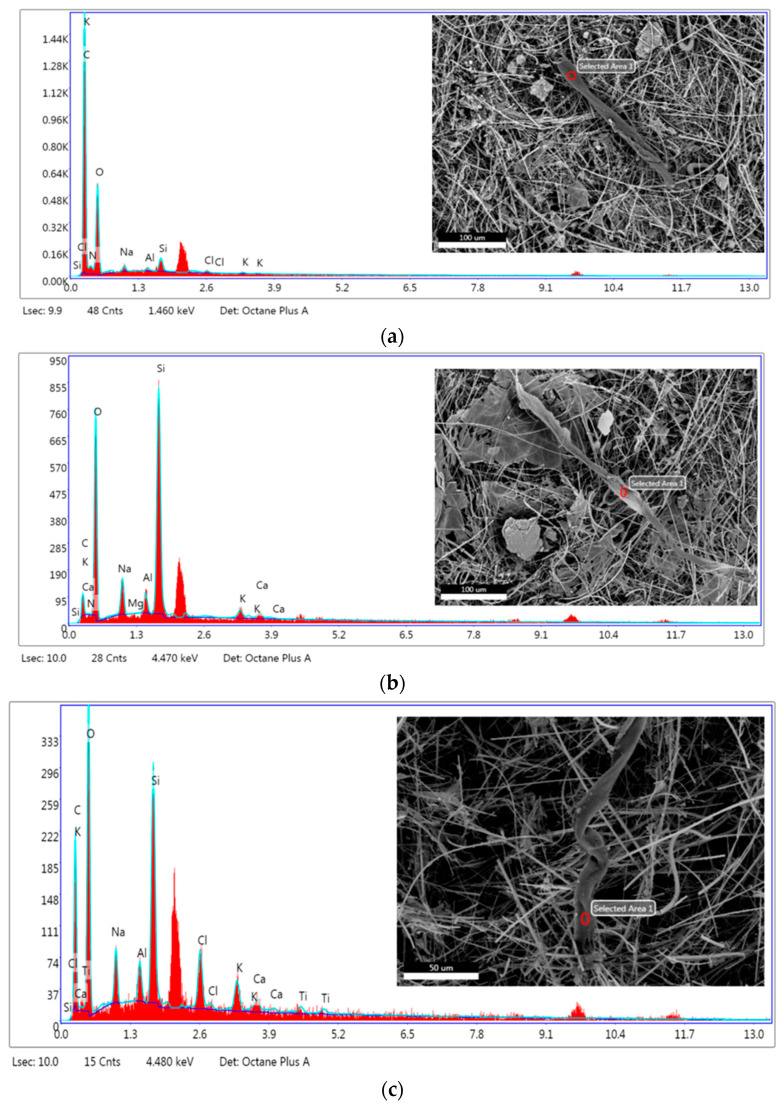
EDX patterns for micro-fragments found in seawaters. (**a**,**b**) Venice-Lido port inlet, (**c**,**d**) Grand Canal at the Rialto Bridge, (**e**,**f**) Saint Marc basin.

**Figure 7 materials-17-01759-f007:**
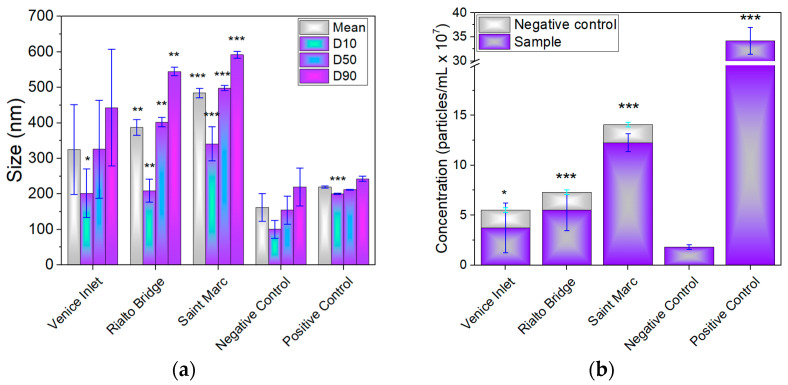
Size (**a**) and concentration of nanoparticles (**b**) measured for seawater samples collected from the Venice Lagoon. D10, D50, and D90 values indicate the percentage of particles under the specific size. * *p* < 0.05, ** *p* < 0.01, and *** *p* < 0.001.

**Table 1 materials-17-01759-t001:** Characteristics of MPs examined in surface water ecosystems.

Study Area	Sampling	Analytical Tool	Characteristics of MPs/NPs	Ref.
Two sites from Wadden Sea (The Netherlands)	5 L Niskin bottle—site A, PP bucket—site B.	Thermal desorption—proton transfer reaction—mass spectrometry (TD-PTR-MS)	PS NPs, PET NPs.	[29]
Saint Marc basin, Rialto Bridge, Venice-Lido port inlet	Bulk, 5 L jerry cans, sampling conducted on 15th June 2022.	Optical microscopy	3.6 ± 1.1 MPs/L, 1.6 ± 1.1 MPs/L, 1 ± 0 MPs/L.	[30]
Banderas Bay, Mexico	Wooplankton net (0.3 m diameter, 333 μm mesh size), floating plastic monitoring conducted from 2016 to 2018.	µ-ATR-FTIR	79% of marine litter were MPs (45% PP, 43% PE). The most represented size class was the 1–2 mm.	[31]
Marine-protected areas from Peru	Bulk method—15 L bucket. Sampling conducted from January to May of 2022.	ATR-FTIR	4.19 ± 2.23 MPs/L (1.60–9.37 MPs/L). The particles were largely composed of fibers (91.6%) of blue color (81.8%). MPs mean size of 1260 µm. Size classes: 46.4% in the range of 1000–5000 µm. 26.2% in the range of 500–1000 µm.	[32]
34 sites from India’s east coast	Manta nets, neuston nets, plankton nets, and bongo nets 750 mL collection jar. Sampling conducted in the monsoon period of August 2022.	Optical microscopy, SEM, and FTIR	43% PE, 42% PP, and 15% PS. Mean: 12 MPs/site. Morphology: 59% fibers, 24% films, 10% fragments, and 7% pellets. Color distribution: 26% white, 16% black, 12% grey 12%, 14% red, 12% blue, 10% yellow, and 10% green.	[33]
16 different stations of the South Yellow Sea	100 L of surface seawater (0–50 cm) was collected in January, April, and August 2018.	ATR-FTIR	High abundance of 6.5 ± 2.1 items/L. 78% of MPs were <500 μm. ~90% were fibers.	[34]
Tuscan coast, Italy	Manta trawl (330 μm mesh size), 24 surface tows. Sampling conducted on November–December 2013 and April–May 2014.	Stereomicroscope FT-IR	1586 MPs, corresponding to 0.16 ± 0.26 mg/m^3^. The most abundant size class is 1–2.5 mm; 81% fragments; PE > 66% and PP 28%.	[35]

**Table 2 materials-17-01759-t002:** Physical and chemical characteristics of seawater collected from the Venice Lagoon.

	Venice-Lido Port Inlet	Rialto Bridge	Saint Marc Basin
pH (pH units)	7.8 ± 0.04	7.7 ± 0.02	7.9 ± 0
Electrical conductivity (mS/cm)	49.5 ± 0.47	48.7 ± 0.1	48.8 ± 0.05
Salinity	32.5 ± 0.15	31.7 ± 0.15	31.8 ± 0.2
Chloride ions (g/L)	20.2 ± 0.95	14.4 ± 0.7	20.1 ± 1.13
Total dissolved solids (g/L)	30.8 ± 0.15	30.0 ± 0.15	30.1 ± 0.15

**Table 3 materials-17-01759-t003:** Elemental composition for the micro-fragments detected via EDX analysis for the samples shown in Figure 6.

Element	Weight (%)					
	Venice-Lido Port Inlet		Grand Canal at the Rialto Bridge		Saint Marc Basin	
	(a)	(b)	(c)	(d)	(e)	(f)
Carbon (C)	62.83	16.45	35.42		31.01	41.85
Nitrogen (N)	1.50	0.03				
Oxygen (O)	32.51	47.78	41.20	55.84	42.32	39.28
Sodium (Na)	1.21	6.62	4.29	5.80	5.32	5.57
Magnesium (Mg)		0.04			0.44	0.63
Aluminum (Al)	0.33	2.30	1.79	1.60	1.69	0.77
Silicon (Si)	1.22	23.86	9.72	33.07	9.83	3.69
Chloride (Cl)	0.21			3.69		0.36
Potassium (K)	0.19	1.57	2.35	3.69	8.68	7.87
Calcium (Ca)		1.35	1.19			
Titanium (Ti)			0.68		0.72	

**Table 4 materials-17-01759-t004:** Estimation of MP and nanoparticle concentration found in investigated seawater samples.

Water Sample	Concentration of MPs (mg/m^3^)	Concentration of Nanoparticles (mg/m^3^)	Total Concentration (mg/m^3^)
Venice-Lido port inlet	67.31	0.66 × 10^−3^	67.311
Grand Canal at the Rialto Bridge	34.93	0.167 × 10^−2^	34.397
Saint Marc basin	256.52	0.727 × 10^−2^	256.532

## Data Availability

Data will be made available on request due to privacy.

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
