# Peer review of "Assessing Microplastics and Nanoparticles in the Surface Seawater of Venice Lagoon—Part I: Methodology of Research"

_materials, 2024, doi:10.3390/ma17081759_

Round 1
Reviewer 1 Report
Comments and Suggestions for Authors
I have carefully read the manuscript entitled “Assessing Microplastics and Nanoparticles in the Surface Seawater of Venice Lagoon. Part I: Workflow Pattern for Methodology of Research” that was submitted to Materials. Microplastics (MPs) and nanoplastics (NPs) are important problems to be solved urgently in environmental science. The aim of this work was to develop a convenient and efficient method to detect and measure MPs and NPs in surface seawater by analyzing the relative abundance of MPs by light microscopy and characterizing them according to their color, shape, and size classification, and estimating the concentration and average value of NPs by nanoparticle tracking analysis techniques. In general, this work is interesting. The following are some minor points that may further improve the work:
1. Larger plastics exhibit a lower rate of photoaging compared to smaller plastics. What is the reason for this discrepancy?
2. What are the reasons for the apparent discrepancies in MPs between the three locations of the Venice Port Inlet, the Rialto Bridge and Saint Marc?
3. Plankton webs are valuable tools for studying marine ecosystems, but why may they not capture the full range of MPs, especially the smaller particles?
4. The abundance of MPs seems to be inversely proportional to the concentration of nanoparticles measured in the Venetian lagoon, which is the specific reason.
5. The estimated concentrations of MPs and NPs in this work appear to be lower than those reported in other reports, which may need to be explained.
6. This paper mentions the lack of standardized methods for assessing and quantifying these pollutants and their health effects, but does not seem to mention the progress of research on standardized methods for assessing and quantifying these pollutants and their health effects, which needs to be further supplemented.
Comments on the Quality of English LanguageMinor editing of English language required
Author Response
Thank you for your comments and suggestions!
In the following, the point-by-point answers to the remarks are detailed.
- Larger plastics exhibit a lower rate of photoaging compared to smaller plastics. What is the reason for this discrepancy?
The text was completed as follows:
“The reason for the discrepancy between larger plastics and a lower rate of photoaging compared to smaller plastics is that the nanoplastics exhibit a specific surface area, which enhances the generation of reactive oxygen species ROS [8] having a higher capacity for forming hydroxyl radicals (.OH) [7]. The degradation of plastic debris under sunlight exposure leads to an increasing in surface roughness, allowing smaller particles to easy leach additives from their composition and adsorb other chemical toxic pollutants from aquatic media, making them more dangerous to the aquatic environment.”
- What are the reasons for the apparent discrepancies in MPs between the three locations of the Venice Port Inlet, the Rialto Bridge and Saint Marc?
The possible reasons for the apparent discrepancies in MPs between the three locations of the Venice Port Inlet, the Rialto Bridge and Saint Marc could be:
- “the circulation and types of currents prevalent in the Venice Lagoon, described as a large-scale cyclonic meander [67].” “It is possible that the stronger currents from the Venice Port Inlet act as traps for MPs compared to the Rialto Bridge and Saint Marc sites, while weaker currents, may potentially facilitate the export of MPs.”
- the activity of harbors, which contribute with MPs into Venice Port Inlet site.
- Plankton webs are valuable tools for studying marine ecosystems, but why may they not capture the full range of MPs, especially the smaller particles?
“Zooplankton with sizes ranging between 200 µm to 200 mm, have the capacity to potentially ingest suitably sized MPs [77], often mistaking them for food in aqueous ecosystems. Zooplankton, such as copepods, krill, and various larval forms of marine animals, are also used as indicators of microplastic pollution in aquatic ecosystems by ingesting microplastic particles present in that environment. They help quantify the level of microplastic contamination in aquatic environments by measuring the number of MPs ingested by zooplankton. Ecotoxicological studies assess how microplastic ingestion affects zooplankton survival, reproduction, growth, and physiology. Also, model studies are conducted with zooplankton samples to further understand the dynamics of microplastic pollution in aquatic ecosystems [78]. Various organisms in aquatic ecosystems, including copepods, mysid shrimps, rotifers, cladocerans, ciliates, and polychaete larvae, are a significant part of the food chain and are often food source for other organisms, including fish, birds and marine mammals. The size of MPs is a significant factor influencing the frequency and quantity of plastics ingested by these organisms. Bermudez and Swarzenski [77] investigated the size relationship between predators and prey in aqueous ecosystems. Thus, for dinoflagellates, other flagellates, ciliates, rotifers and copepods, and cladocerans and meroplankton larvae, the size ratios with their prey are 1:1, 3:1, 8:1, 18:1, and approximately 50:1, respectively. In addition, the study notes a difference in prey selectivity between filter feeders and raptorial-interception feeders. Filter feeders tend to prefer relatively smaller prey, while raptorial-interception feeders favor larger prey.”
“However, planktonic web is not able to capture the full range of MPs, especially the smaller particles. It was reported that planktonic copepods, which show a range of prey’s sizes from 200 µm to 2000 µm, can reject 80% of the MPs due to the feeding-current and sensorial mechanism, independent of the type of polymer, morphology, and presence of biofilms [76]. Also, the residence time for the MPs in copepods is short, them being swiftly expelled through fecal pellets.”
- The abundance of MPs seems to be inversely proportional to the concentration of nanoparticles measured in the Venetian lagoon, which is the specific reason.
Although the high abundance of MPs was recorded in Venice Port Inlet site, it was expected that the concentration of nanoparticles to be high. Our results showed a concentration of 3.71x107 particles/mL, which is low than that recorded for Saint Mark and Rialto Bridge sites. This discrepancy could be explained by the low rate of fragmentation of MPs and low ageing of MPs.
- The estimated concentrations of MPs and NPs in this work appear to be lower than those reported in other reports, which may need to be explained.
“The limitations of this methodology are attributed to the rejection of MPs due to mesh sizes ranging from 32 μm down to 1 μm, potential interferences from co-existing colloids and fragments during NTA technique [84], and the high probability of missing small and transparent MPs during the optical microscopy analysis.”
Our estimated concentrations for MPs and nanoparticles were lower than other reported data, due to the sampling method, detection technique used, and methodology applied.
- This paper mentions the lack of standardized methods for assessing and quantifying these pollutants and their health effects, but does not seem to mention the progress of research on standardized methods for assessing and quantifying these pollutants and their health effects, which needs to be further supplemented.
“The progress of research on standardized methods for assessing and quantifying these pollutants and their health effects has made significant progress over the years, driven by the increasing recognition of the impact of pollutants on human health and the environment. The key advancements in this field are illustrated by reports elaborated by the Joint Group of Experts on the Scientific Aspects of Marine Environmental Protection (GESAMP) [51],[52], the Joint Research Center (JRC) [53], and the National Oceanic and Atmospheric Administration's (NOAA) [54], which aim to standardize methods for sampling, analysis, and reporting of MPs data. Also, interlaboratory comparison studies were organized with the purpose of predetermining the recovery rates of spiked MPs in wastewater and sludge sample [55], identifying the type and quantity of MPs [56], generating MPs in standard sample bottles [57], and developing reference materials for method validation in microplastic analysis [58].”
Minor editing of English language required
The manuscript was carefully checked for the accuracy in English language.
Reviewer 2 Report
Comments and Suggestions for Authors
The research was well conducted and reported.
Minor corrections of the English language are required.
For example (not exhaustive): line 475 « Our results suggest a less pollution in the Venice Lagoon.» maybe ‘lower’ instead of ‘less’? In comparison with what?
Comments on the Quality of English Languagenone
Author Response
Thank you for evaluating out manuscript!
Minor corrections of the English language are required.
The manuscript was carefully checked for the accuracy in English language.
For example (not exhaustive): line 475 « Our results suggest a less pollution in the Venice Lagoon.» maybe ‘lower’ instead of ‘less’? In comparison with what?
Thank you for your comment. We decided to delete this phase because it was mention in the text that:
“Our estimated concentrations for MPs and nanoparticles were lower than other reported data, possibly due to the sampling method, detection technique used, and methodology applied.”
Reviewer 3 Report
Comments and Suggestions for Authors
Dear Authors
First of all, let me congratulate you on such interesting work. The proposed methodology seems to be capable of detecting all types of wastes related with MP and nanoplastics.
The restrictions presented in the division of particles using filters of different sizes should be improved because the rejected amount of the systems seems to be high.
You present an ANOVA treatment, but you've not explained the results obtained from the statistical point of view. Please, consider to add because in a work such this it is very important to be sure that the examination is accurate enough to extract different conclusions.
Flow patterns are not very well described. MAybe it would be interesting, being the part 1, to change the title of the article if you cannot talk about the flowing patterns detected. The very summarized explanation is not more than a mere comment. Please, consider that fact in revisiong the article, please.
In equation 1 (page 7, line 231) the meaning of each parameter of the Einstein-Stokes mathematical expression, D is the diffusion coefficient. The average distance, in the D value calculation, is divided by 2t (twice the time used for the displacement). Please, consider changing it in the text and take into account if you calculate because D, should be calculated specifically using the time as parameter.
On page 7, line 242, PS acronym should be explained in more detail (polystyrene nanoparticles) because it is used as a standard to calibrate the nanosize measurement. Please, aport more details about this.
Many thanks
Author Response
Thank you for your comments and suggestions!
The restrictions presented in the division of particles using filters of different sizes should be improved because the rejected amount of the systems seems to be high.
Thank you for your comment!
In this paper we used five size classes for collecting of MPs: > 1 mm, 1000–250 μm, 250–125 μm, 125–90 μm, and 90–32 μm for collecting of MPs. According to the definition of microplastics, all plastic particles with sizes up to 1 μm should be retained. Considering that the abundance of MPs increases as the mesh size decreases, a high number of particles was rejected.
We corrected the text:
“The limitations of this methodology are attributed to the rejection of MPs due to mesh sizes ranging from 32 μm down to 1 μm, potential interferences from co-existing colloids and fragments during NTA technique [84], and the high probability of missing small and transparent MPs during the optical microscopy analysis.”
For further researches we improved the processing protocol and analyzed plastic particles up to 20 μm, and subsequently down to 1 μm.
You present an ANOVA treatment, but you've not explained the results obtained from the statistical point of view. Please, consider to add because in a work such this it is very important to be sure that the examination is accurate enough to extract different conclusions.
The ANOVA analysis did not reveal a statistically significant difference among the number and color of MPs detected in the Venice Lagoon (p > 0.05). Statistically significant differences between the sample locations were obtained for the size classes of 125-90 µm (p = 0.01) and 90-32 µm (p = 0.02).
Flow patterns are not very well described. MAybe it would be interesting, being the part 1, to change the title of the article if you cannot talk about the flowing patterns detected. The very summarized explanation is not more than a mere comment. Please, consider that fact in revisiong the article, please.
We changed the tiple of manuscript in “Assessing Microplastics and Nanoparticles in the Surface Seawater of Venice Lagoon. Part I: Methodology of Research”.
In equation 1 (page 7, line 231) the meaning of each parameter of the Einstein-Stokes mathematical expression, D is the diffusion coefficient. The average distance, in the D value calculation, is divided by 2t (twice the time used for the displacement). Please, consider changing it in the text and take into account if you calculate because D, should be calculated specifically using the time as parameter.
In equation 1, we changed the meaning of “D” as diffusion coefficient.
On page 7, line 242, PS acronym should be explained in more detail (polystyrene nanoparticles) because it is used as a standard to calibrate the nanosize measurement. Please, aport more details about this.
The text was completed with the certified diameter 204±3.1 nm for PS NPs and the dilution solution, as follows:
“For positive control in nanoparticle analysis, polystyrene nanoparticles (PS NPs) (standard latex NTA4089, Malvern Panalytical, UK, certified diameter 204±3.1 nm) were diluted 20x in medical grade saline solution of 9 mg/mL NaCl.”